# Automation of Modeling and Calibration of Integrated Preparative Protein Chromatography Systems

Simon Tallvod, Niklas Andersson  and Bernt Nilsson *

Department of Chemical Engineering, Lund University, 22100 Lund, Sweden;
simon.tallvod@chemeng.lth.se (S.T.); niklas.andersson@chemeng.lth.se (N.A.)
* Correspondence: bernt.nilsson@chemeng.lth.se; Tel.: +46-46-222-8088

**Abstract:** With the increasing global demand for precise and efficient pharmaceuticals and the biopharma industry moving towards Industry 4.0, the need for advanced process integration, automation, and modeling has increased as well. In this work, a method for automatic modeling and calibration of an integrated preparative chromatographic system for pharmaceutical development and production is presented. Based on a user-defined system description, a system model was automatically generated and then calibrated using a sequence of experiments. The system description and model was implemented in the Python-based preparative chromatography control software Orbit.

**Keywords:** model calibration; digital twin; preparative chromatography; chromatography modeling; integrated continuous processes; process automation

## 1. Introduction

The biopharmaceutical market is constantly changing with an increasing demand for precise and efficient drugs at affordable prices and with global access. There is high pressure to reduce costs for the development and manufacture of biologics [1,2].

To increase the productivity in biopharmaceutical development and production, two emerging technologies are key: (i) the introduction of integrated continuous biomanufacturing, which leads to fewer buffer and storage tanks, smaller equipment sizes, and full automation of the operation as well as increased process flexibility [2,3]; (ii) the development of digital solutions (i.e., digitalization), which help to save time and costs in process development as well as increase the number of possibilities for process control and quality assurance [4].

To facilitate the development, design, and optimization of integrated downstream processes, mathematical modeling and model-based studies become important. Model-based studies of chromatography processes for performance studies, design issues, optimization, robustness, and control have been published [5–8] as well as for continuous downstream processes [9,10].

Mathematical modeling of chromatography is becoming an integrated part of process development in the biopharmaceutical industry. Dynamic models are used to study the separation of a target protein from impurities and how the performance changes with the design and operation parameters. Several different models have been published, e.g., models for ion exchange, hydrophobic interaction, reversed phase, mixed mode, and affinity [11]. It is essential that the model can capture different concentrations, load volumes, and elution gradients in a chromatographic separation process. Calibration (i.e., the estimation of the parameter values of the mathematical model to fit its response to experimental responses) is a vital part of modeling. In practical cases, this amounts to fitting the multiple model–experiment responses after each other, capturing a wide range of operational conditions for a design space. To develop a detailed and accurate preparative chromatography model, validated to well-chosen experiments, for model-based

studies requires both competence and resources. The selection of a proper model structure capable of capturing the system's behavior with minimum model complexity and number of parameters together with a set of calibration experiments demands a skilled model developer.

In order to speed up the implementation of model-based design in the development of complex manufacturing systems in the biopharmaceutical industry, process modeling, including parameter estimation and model calibration, needs to become fast, efficient, and robust. In addition, when moving towards an industry with more autonomous production and interconnected processes—a paradigm shift referred to as Industry 4.0—the need arises for realistic models of the process states [12] for use in technologies such as digital twins [13,14] and advanced control strategies such as model predictive control [15]. It is noteworthy that these models do not need to be mechanistic, such as the models used in this work, but could be data-driven as well [16].

This work addressed the abovementioned issues by presenting a framework for the automatic generation of a model structure of a laboratory chromatography setup together with an associated automatic procedure to perform a sequence of experiments for parameter estimations. The aim was to illustrate how to enhance and speed up the modeling of chromatography processes, how to simplify model development and model maintenance, and to reduce the number of resources needed to acquire an accurate model. The results can be used for model-based in silico studies but also directly integrated into the chromatography control system. This concept was implemented in Orbit, a real-time control system. Based on a setup configuration, Orbit automatically generates a model structure of the configuration. Orbit performs a set of experiments both on the physical setup and on the digital model and finds proper model parameters. An ion exchange example, with gradient elution, was chosen as a case study to illustrate the concept of automated modeling.

The paper first presents some fundamental aspects of the mathematical modeling needed to explain the model calibration of the case study. The following sections detail the experiments performed and the software implementation. The main results are a presentation of how the model generation is performed in Orbit and how the calibration procedure is performed with minimal user interaction.

## 2. Theory

### 2.1. Mathematical Modeling

The automatic generation of a model structure of an experimental setup assumes that each part of the setup has an individual mathematical representation. All components of the setup—buffers, tubes, pumps, mixers, valves, chromatography columns, and detectors—will have a parameterized mathematical model structure. The dynamic behavior of the concentration of different components in the mobile phase is modeled by three different model structures: (i) a well-stirred tank, (ii) a tube, and (iii) a porous packed bed (with or without adsorption). The fluid dynamics are not captured in the model structures, and the flow rate is assumed to be changed instantaneously by the pump in the flow path [17,18].

The concentration of a component, $c_i$, in some objects (e.g., mixers, valves, and detectors) is described by well-stirred tank models:

$$\frac{dc_i}{dt} = \frac{1}{V_k} \sum_{j=1}^{m} F_j \left( c_{in,i,j} - c_i \right) \tag{1}$$

where $V_k$ is the volume; $m$ is the number of inlets; $F_j$ is the flow rate. Index $k$ denotes the unit and index $j$, the inlet.

The concentrations in the tubes are modeled by a dispersive–convective description

$$\frac{\partial c_i}{\partial t} = D_{ax,t} \frac{\partial^2 c_i}{\partial z^2} - \frac{F_k}{A_k} \frac{\partial c_i}{\partial z} \tag{2}$$

where $D_{ax,t}$ is the tube dispersion coefficient; $F_k$ is the flow rate; $A_k$ is the cross-sectional area of the tube with Danckwerts boundary conditions, i.e., with a flux condition at the inlet and no flux at the outlet. Note that, in general, the flow rate may change direction, resulting in a corresponding change in the model structure.

Packed beds are also described by dispersive–convective descriptions, similar to tubes, but the parameters have a slightly different meaning, capturing the porous packing properties:

$$\frac{\partial c_i}{\partial t} = D_{ax,c}\frac{\partial^2 c_i}{\partial z^2} - \frac{F_k}{\varepsilon_{T,i}A_{c,k}}\frac{\partial c_i}{\partial z} - \frac{1-\varepsilon_c}{\varepsilon_{T,i}}r_{ads,i} \tag{3}$$

where $D_{ax,c}$ is the packed bed dispersion coefficient; $A_{c,k}$ is the column's cross-sectional area; $\varepsilon_c$ is the void volume ratio; $\varepsilon_{T,i} = \varepsilon_c + (1-\varepsilon_c)\varepsilon_i$ is the component-specific porosity with Danckwerts boundary conditions as in the tube model.

Ion exchange adsorption is described by the classic steric mass action (SMA) model, with a salt-dependent elution and competitive protein adsorption [11,19–22]:

$$\frac{\partial q_i}{\partial t} = r_{ads,i} = k_{kin,i}\left[K_{eq,i}c_i\left(\Lambda - \sum_{j=1}^{n_{comp}}(\nu_j - \sigma_j)q_j\right)^{\nu_i} - c_s^{\nu_i}q_i\right] \tag{4}$$

where $q$ is the concentration of the adsorbed component; $c_s$ is the salt concentration; $\Lambda$ is the ligand density; $\nu$ is the characteristic charge; $\sigma$ is the shielding factor; $K_{eq}$ is the equilibrium parameter; finally, $k_{kin}$ is the adsorption kinetic parameter. The expression can be reformulated as:

$$\frac{\partial q_i}{\partial t} = r_{ads,i} = k_{kin,i}\left[H_{0,i}c_i\left(1 - \sum_{j=1}^{n_{comp}}\frac{q_j}{q_{max,j}}\right)^{\nu_i} - c_s^{\nu_i}q_i\right] \tag{5}$$

with the parameters:

$$H_{0,i} = K_{eq,i}\Lambda^{\nu_i} \tag{6}$$

$$q_{max,i} = \frac{\Lambda}{\nu_i - \sigma_i} \tag{7}$$

where $H_{0,i}$ is the Henry equilibrium constant, and $q_{max,i}$ is the column binding capacity.

The model can handle small variations in pH by adjusting the characteristic charge and equilibrium constant with the adjustment parameters $a$ and $b$ [23–25].

$$H_{0,i}(pH) = H_{0,i,pH_{ref}}e^{a_{pH,1}\Delta pH + a_{pH,2}\Delta pH^2} \tag{8}$$

$$\nu_i(pH) = \nu_{i,pH_{ref}} + b_{pH}\Delta pH \tag{9}$$

A complete model structure for the simulation of gradient elution has many parameters, some specific to the setup and configuration and others specific to adsorption. To determine all of the model parameter values, a sequence of experiments with corresponding parameter estimations needs to be performed [23,25–33].

### 2.2. Yamamoto Method

A way to estimate $\nu_i$ and $H_{0,i}$, as shown in Equation (5), from linear gradient experiments is the Yamamoto method [25,34].

$$\log(GH) = (\nu_i + 1) \cdot \log(c_s) - \log(H_{0,i} \cdot (\nu_i + 1)) \tag{10}$$

$$g = \frac{c_{G,final} - c_{G,initial}}{V_G} \tag{11}$$

$$GH = g(V_{col} - \varepsilon_c V_{col}) \tag{12}$$

where $GH$ is the normalized gradient slope; $c_{s,i}$ is the salt concentration at the maximum of an eluted protein peak; $c_{G,initial}$ and $c_{G,final}$ are the initial and final salt concentrations

of the gradient; $V_G$ is the gradient volume; $V_{col}$ is the column interstitial volume; $\varepsilon_c$ is the column void. The parameters are found by means of linear regression and the expressions above.

## 3. Materials and Methods

### 3.1. Materials

Two buffers were used in this work. The first buffer, Buffer A, contained 20 mM sodium phosphate, and it was used for column equilibration, loading, washing, and for general flow. The second, Buffer B, contained 20 mM sodium phosphate as well as 0.5 M sodium chloride, and it was used for elution and column regeneration. Both buffers had a pH of 7.0. The first of the two samples used was a mixture of 1 g/L blue dextran 2000 (GE Healthcare, Uppsala, Sweden) and 5 g/L acetone, and the second sample was 1 g/L cytochrome c from equine heart (Sigma–Aldrich, St. Louis, MO, USA). Both samples were prepared using Buffer A. The chromatographic column used was a 1 mL HiTrap Capto SP ImpRes (Cytiva, Uppsala, Sweden) with a bed height of 25 mm [35].

### 3.2. Experimental Setup

An ÄKTA Pure 25 preparative chromatographic system (Cytiva, Uppsala, Sweden) was used, and it consisted of a two-channel gradient pump with one inlet valve per channel, a loop valve with two super loops, an injection valve, a column valve, a UV detector, a conductivity detector, a pH detector, and an outlet valve. The system was configured as seen in Figure 1. The units were connected using PEEK tubing with an inner diameter of 0.75 mm, and the total tube length was approximately 3.5 m between the pump, through the protein sample super loop, and the outlet valve. Buffer A was connected to the A side of the pump and Buffer B to the B side. The two samples were loaded into one super loop each.

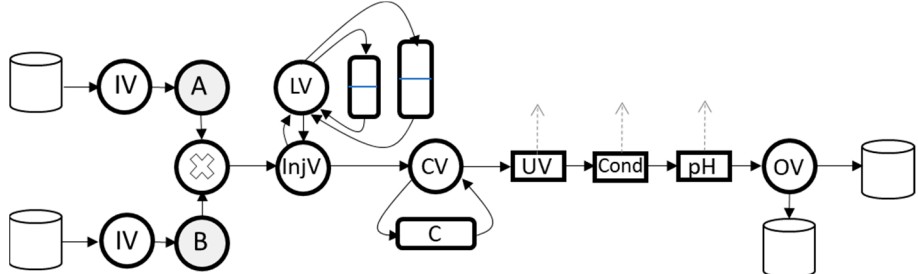

**Figure 1.** Illustration of the experimental setup configuration. The system consisted of buffer bottles connected to inlet valves (IVs) via pumps (A/B); an injection valve (InjV); a loop valve (LV) with super loops; a column valve (CV) with column (C); UV, conductivity, and pH sensors; an outlet valve (OV).

### 3.3. Experiments

The calibration procedure used a sequence of four types of experiments: by-pass (I), packed bed (II), linear gradient (III), and overloading (IV) (see Table 1).

**Table 1.** A summary of the calibration steps performed and their associated experiments, parameters, and objective functions.

| Calibration Step | Experiment | Parameters | Objective Function (SSE) |
|:---:|:---:|:---:|:---:|
| I | By-pass | Tube length and UV sensor volume | Retention time and UV |
| II | Packed bed | Column void and porosity | UV |
| III | Linear gradient | $v_i$ and $H_{0,i}$ (Yamamoto method) and $k_{kin,i}$ | $\log(GH)$, $v_i$ and $H_{0,i}$, UV $(k_{kin,i})$ |
| IV | Overloading | $q_{max,i}$ | UV |

### 3.3.1. By-Pass Experiment (I)

In order to calibrate the dead volume of the system, 0.5 mL of the 1 g/L protein sample was injected into the system with the column by-passed. The buffer used was Buffer A, and the flow rate was 2 mL/min. The signal from the UV detector was used for calibration and for the determination of the protein's extinction coefficient.

### 3.3.2. Packed Bed Experiment (II)

To calibrate the void and porosity of the column, the column was equilibrated with 1 column volume (CV) of Buffer A after which 0.5 mL of the sample containing 1 g/L blue dextran and 5 g/L acetone were injected into the system and through the column. The flow rate was 1 mL/min. The signal used for calibration was taken from the UV detector.

### 3.3.3. Linear Gradient Experiment (III)

The experiment was implemented by cleaning and equilibrating the column with Buffer B followed by Buffer A for 3 CV and 4 CV, respectively. Then, 1 mL of the 1 g/L protein sample was injected into the column followed by a 2 CV wash with Buffer A. A linear gradient was applied to the column after which the column was again cleaned and equilibrated with Buffer B and Buffer A for 3 CV and 5 CV, respectively. The signals used for calibration were UV and conductivity.

### 3.3.4. Overloading Experiment (IV)

The column was equilibrated with Buffer A for 3 CV after which 20 CV of the 1 g/L protein sample was loaded into the column. This was followed by a 2 CV wash with Buffer A and a linear gradient between 0 and 100% of Buffer B for 30 CV. The column was then cleaned and equilibrated with Buffer B for 2 CV followed by 3 CV of Buffer A. The UV signal was used for calibration.

### *3.4. Orbit*

### 3.4.1. External Controller for Physical Experiments

In order to control and simulate the chromatography system and process, a software called Orbit was used [36]. Orbit is a Python-based software, developed at the Department of Chemical Engineering at Lund University, that allows for control of several brands of chromatography systems, interfacing of auxiliary equipment and sensors, sequencing, and automation as well as simulation of chromatographic processes. Orbit communicates with the ÄKTA system via an application programming interface (API) to the software Unicorn (Cytiva, Uppsala, Sweden) which, in turn, controls the system. There are other commercially available pieces of software for chromatographic modeling and simulation (e.g., CADET [37]), which have extensive libraries of chromatographic models and strategies for calibration and process design. However, the strength of Orbit is that it can be used to control chromatographic systems as well as to simulate them.

To run a process, the user defines the system by specifying which system subunits are present in the system, how the subunits are connected, and the approximate lengths of the tubes that connect them. If the process is to be simulated, the buffers and samples need to be defined as well. The user defines a run by specifying a sequence of phases that can be run either experimentally or as a simulation. Examples of phases are a wash step, a gradient elution, or a cleaning step. Phases consists of specific commands that are sent to the chromatography system, e.g., set the position of a valve, set the gradient buffer percentage, or change the flow rate.

### 3.4.2. Automatically Generated Simulator

Once the user has specified the system and sequence of phases, the sequence can be executed, and the system run. Besides running the actual physical system, Orbit has a simulator that can simulate it instead. To achieve this, Orbit has a unit library where all system subunits are defined with mathematical models and default parameter values.

Mixers, valves, and detectors are modeled using Equation (1); tubes are modeled using Equation (2); ion exchange columns are modeled using Equation (5). The system model is automatically generated from this library and the user-defined system description. The system model is automatically updated according to the user-specified control sequence. The two main models used were a well-stirred tank model and a dispersive–convective model. The tank model was used for the valves, pumps, and detectors, while the dispersive–convective model was used for the tubes and chromatographic columns. The convective term was discretized using a two-point backward approximation, while the dispersion term used a three-point central approximation. The grid sizes of the tubes and columns are user-definable, and the nominal grid size in the tubes is approximately one grid point per five centimeters of tube, while the column is discretized using 50 grid points. This gives a residence time of 0.67 s per grid point at a flow rate of 2 mL/min in the tubes. The model was simulated in Python using the backward differentiation formula (BDF) method with a sparse Jacobian matrix [38]. The volumetric flow rate was set in the pump object and the velocity in each subsequent unit was calculated according to their respective cross-sectional area. The simulation steps through the phases, updates the flow rate and sets the relationship between units depending on the current flow path, changing the flow rate of units outside of the flow path to zero.

### 3.4.3. Model Calibration

The model calibration was performed by comparing the experimental results with the simulated runs and minimizing an objective function with respect to a specific model parameter. In order to compute objective functions, peaks in the chromatograms were automatically detected using differentiation of the signals using the method described in [39]. Two kinds of objective functions were used: the squared difference in retention time of experimental and simulated peaks and the sum of squared differences (SSD) between the experimental and simulated chromatograms around a peak. The minimization was performed using Python and the derivative-free Nelder–Mead and Powell methods with bounds at or close to zero, depending on the decision variable [40]. There are of course other methods of calculating the objective function than SSD [41]; however, in this study the SSD was deemed sufficient.

## 4. Results and Discussion

Based on the description of the system configuration in Orbit, an implementation of the mathematical model was automatically generated that could then be simulated. The user must specify the system configuration, i.e., which units are present, how they are connected, and the length of tubing between them. If the user wishes, they may change additional physical parameters (e.g., internal unit volumes), otherwise those parameters are set to their nominal values which are predefined in the Orbit unit library. If the user has supplied reasonable estimates of the tube lengths, the simulator will yield a result that is close to a real run. This will form the basis of the succeeding calibration steps. When ion exchange chromatography is to be modeled and calibrated, the user also needs to specify buffers that will ensure that the target protein will bind to the column at zero percent elution buffer and elute at 100 percent elution buffer.

### 4.1. By-Pass Modeling and Calibration

The aim of this calibration step was to model the non-column dead time of the system. The calibration was performed by injecting 0.5 mL of the protein sample into the flow path with a flow rate of 2 mL/min. The experiment was then simulated using the exact same set of instructions as the physical experiment, and the user's predefined tube lengths were used for the nominal model response. Before calibration, an experimental run was used to estimate the extinction coefficient of the target protein. The model calibration was performed minimizing a weighted sum of the squared difference in retention time and the sum of squared differences between chromatograms using two decision variables: the

length of the tube going into the UV detector and the internal volume of the UV detector. The results of this calibration step are shown in Figure 2. It is clear that the simulated nominal case was not a perfect fit with regards to retention time, and after calibration, the simulation approached the experimental value.

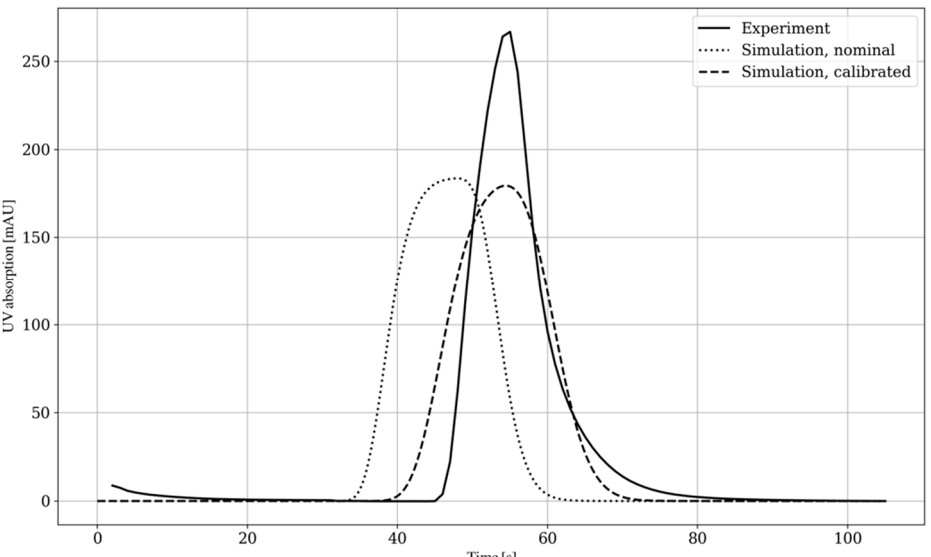

**Figure 2.** Calibration of the system's dead volume without a column: experimental result (solid), nominal model response (dotted); calibrated model response (dashed).

The grid density in a tube was set to a factor of 0.2 grid points per centimeter of tube, or so that the residence time in each grid point was 0.67 s at a flow rate of 2 mL/min and with a tube inner diameter of 0.75 mm. Although not part of the automatic calibration procedure, the tube model grid's resolution greatly influences the shape of the simulated peaks via numerical dispersion. Using only a few grid points in the tube discretization results in broad peaks, with the axial dispersion behavior of the model becoming overshadowed. This makes the grid density in essence take over the role of the dispersive part of the model, making it necessary to set the axial dispersion coefficient, $D_{ax}$, to a low value to avoid too much widening of the peaks. Given the total length of tubes in the system, an increased grid density quickly increased the size of the problem, resulting in a longer computation time. This could be a problem if, for example, the simulator was to be used for time-critical decision making during a run. If, however, the axial dispersion in the tubes is of interest, a finer grid must be used, and a more realistic $D_{ax}$ has to be assessed. This can be performed in an additional calibration step. There are more refined ways of estimating the propagation of a species in a tube than the 1D implementation used in this work, e.g., to better capture radial dispersion, higher dimension convective–diffusion models need to be used [42]. Using the length of the tube entering the UV detector and the detector's internal volume as the two decision variables can be seen as compensating for the model's error by putting all the inaccuracies in these two parameters. Lumping together the errors in the last units before the detector makes for a more flexible setup, since no knowledge of the upstream system is needed before calibration. This also implies that the system has to have a greater volume and length than is first captured in the model. Since this method only changes the length and volume of the two units, the calibration can only make them so small before they become negative, i.e., it can only somewhat compensate if the model turns out to be slower than the experiment. If this is the case, the user must change the specified tube lengths. An alternative to this method is to multiply each tube length by a global factor and use that factor as a decision variable instead. However, this method was not explored further in this work.

*4.2. Packed Bed Modeling and Calibration*

This calibration step was performed in order to find the void and porosity of the chromatographic column, and it was conducted by injecting 0.5 mL of the blue dextran and acetone sample into the column at a flow rate of 1 mL/min. The model calibration was then performed in two steps. First, the column void was found by minimizing the squared difference between the retention times of the blue dextran peaks of the experiment and simulation, using the column void as a decision variable. Secondly, the column porosity was found in the same manner, this time with the acetone peaks and with the porosity as the decision variable instead. It is vital that the two species of tracers do not interact with the stationary phase and only pass through with the mobile phase. Blue dextran was chosen as it only moves in the void between particles and acetone was chosen as it also passes into the pores, thereby experiencing the total void of the column. Since both species are visible by the UV sensor, the calibration could be performed without having to compensate for the offset between sensors that would be needed if a salt was used instead of acetone. A possible problem with blue dextran is that it may lead to tailing peaks, which could be avoided using latex particles [17]. However, this was not further explored in this work. A requirement for this calibration step is that the column volume is supplied by the user. The axial dispersion in the column was set to a value corresponding to a Peclet number of 0.5 at a flow rate of 1 mL/min, according to the method described in [43].

It would be possible to perform an experiment to calibrate the specific porosity of the protein; however, this would require an experimental design that guarantees no adsorption of the protein in the column, which is nontrivial, especially if the protein in question is not well known beforehand. Of course, the user may enter their own value of specific porosity if that is known.

*4.3. Linear Gradient Adsorption Modeling and Calibration*

This calibration step consists of two separate partial steps. First, $\nu$ and $H_{0,i}$ are estimated using linear gradient experiments, and then $k_{kin,i}$ is calibrated using the same experiments.

Three linear gradient experiments were performed in order to estimate the $\nu$ and $H_{0,i}$ of the model using the Yamamoto method as described in Equations (10)–(12) above. The three experiments used successively flatter linear gradients: 0–100% Buffer B for 10 CV; 20–80% Buffer B for 14 CV; 30–70% Buffer B for 18 CV. After estimating $\nu$ and $H_{0,i}$, the model parameter $k_{kin,i}$ was calibrated by comparing the second linear gradient experiment with a simulated run, using the sum of squared errors between the two chromatograms around the elution peak as a minimization objective. The three gradient experiments and the calibrated model responses are shown in Figure 3. It is clear from the figure that the calibrated model fits the experiments well for the three gradient experiments. It can also be seen that the changes in the gradient incline and start and end salt concentrations makes the protein elute at slightly different times, which is appropriate when using the Yamamoto method.

When choosing the gradients some care needs to be taken. The target protein must elute during the gradient and the three gradients need to have different slopes. The simplest way to achieve this is to require the user to make sure that the target protein binds to the column at no or low concentrations of elution buffer and that it elutes at high concentrations. Then, with no prior knowledge of where during the gradient the protein elutes, the three gradients can each be run between 0 and 100% elution buffer, and only made longer each run. This can be done automatically. Another way is to run one gradient between 0 and 100% and see at which salt concentration the protein elutes and then pivot the remaining gradients around that value. Of course, this may not be as robust as the former method, but it may save some time during calibration.

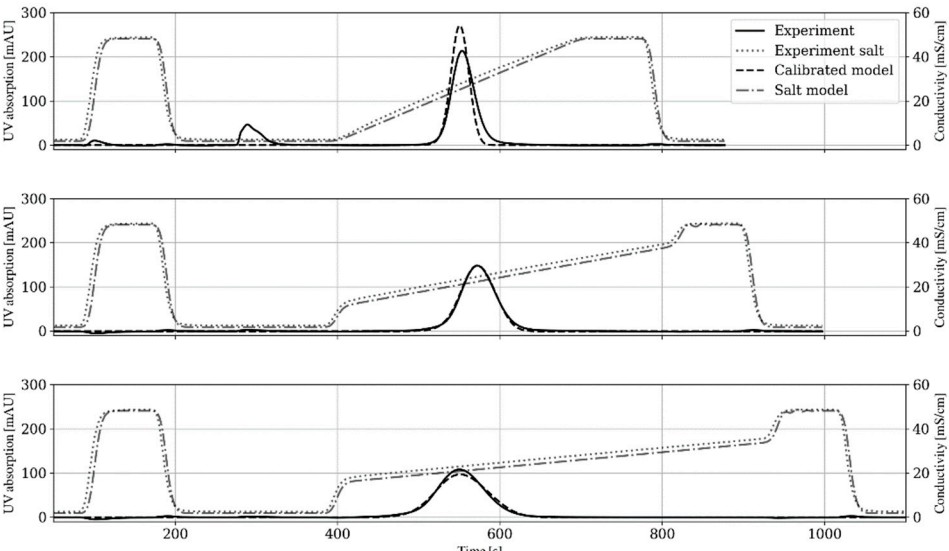

**Figure 3.** Three gradient experiment results with calibrated simulation. The gradient was successively made flatter and longer for each experiment, making the sample protein elute at slightly different times.

If one wishes to capture the pH dependence of the adsorption parameters, the series of gradient experiments can be redone with buffers at a different pH, and then calibration can be performed using Equations (8) and (9).

### 4.4. Overloaded Adsorption Modeling and Calibration

To capture the overloading behavior of the adsorption model, the column binding capacity, $q_{max}$, was calibrated by running an overloading experiment where the column was loaded with 20 CV of the protein sample and then eluted, producing a characteristic fronting peak. The experimental result was compared to a simulated run, and $q_{max}$ was found by minimizing the sum of squared errors between the two chromatograms around the peaks. The experimental result and calibrated model response are shown in Figure 4. The figure clearly illustrates that the model captured nonlinear adsorption and fit the experimental data in a satisfactory way. To guarantee that overloading occurs in this experiment, the user needs to provide a loading volume that ensures a nonlinear adsorption behavior in the column. In this work, a loading factor of 25% of the column's capacity was used [35].

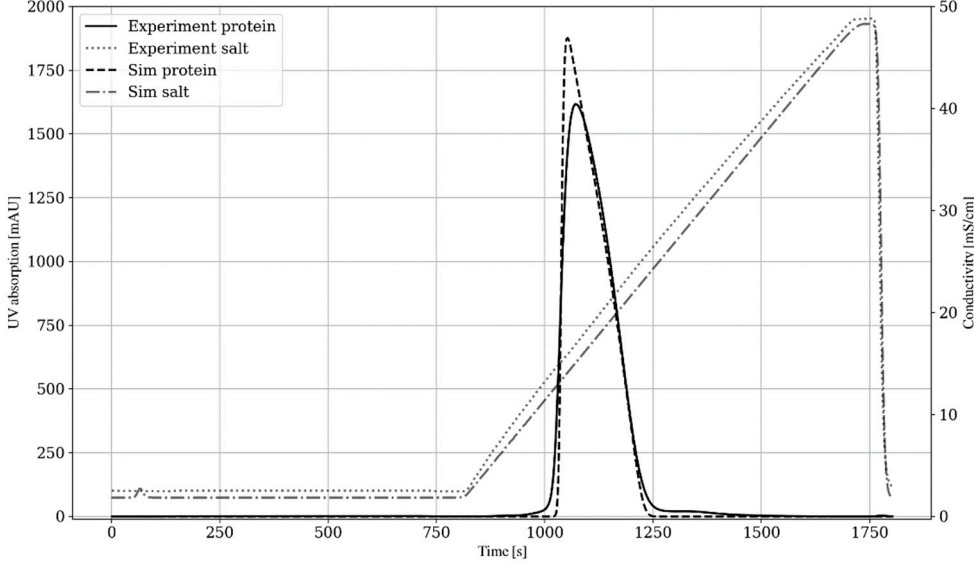

**Figure 4.** Overloaded experiment with a calibrated simulation.

*4.5. General Discussion*

In this work, only one component was modeled and calibrated. If more components need to be simulated, one can easily add more experiments to calibrate each additional component separately. If it is not possible to run an experiment for each component (e.g., if no pure samples of the constituent proteins are available), the calibration method needs to be modified to handle a mix of components. This could be achieved by designing a gradient experiment that ensures that the components elute as separate peaks and then making the gradient flatter in subsequent experiments as discussed above.

One problem that arises when running the calibration automatically is verifying that the calibration was successful and that it sufficiently described the modeled behavior. This validation can be performed as an extra step after the calibration by adding at least one experiment. In the case of cation chromatography, as in this work, an additional linear gradient experiment could be carried out with a gradient somewhere between the gradients in the calibration; furthermore, the column could be slightly overloaded so as to excite the overloading behavior in the model. To quantify if the model was adequately calibrated one could, for example, calculate the coefficient of determination, or $r^2$, between the experimental data and a simulated run, and then it is up to the user to determine what constitutes a good enough fit. Depending on what the model is to be used for, more advanced strategies for determining fit could be used as discussed in [41].

## 5. Conclusions

The work presented in this paper described a proof of concept for automatically generating and calibrating a model of a chromatographic system. The method requires that a user defines the physical system using a predefined library of units in the control software Orbit. Orbit then generates a system model structure that can be simulated or further calibrated. The calibration procedure consists of several steps, each with one or several experiments, designed to calibrate one or more parameters in the model: a by-pass experiment, a packed bed experiment, three linear gradient experiments, and an overloading experiment. The experiments are run automatically and in sequence without manual oversight as is the subsequent model calibration.

As a proof of concept, this work presented the successful application of the automatic modeling and calibration of a chromatography system. The calibration method in this work was tested on a cation chromatography case, but it could easily be customized for other chromatography types by extending the library of chromatographic models in Orbit. The calibrated system model can be used to facilitate and speed up process development and process control in the pharmaceutical industry, moving towards more advanced system representations such as digital twins.

**Author Contributions:** Conceptualization, S.T. and B.N.; Investigation, S.T.; Methodology, S.T.; Software, S.T. and N.A.; Supervision, N.A. and B.N.; Writing—original draft, S.T. and B.N.; Writing—review and editing, S.T. All authors have read and agreed to the published version of the manuscript.

**Funding:** The authors acknowledge that this research was part of the *AutoPilot* project, which is funded by the Swedish Agency of Innovation, VINNOVA (grant number: 2019-05314).

**Institutional Review Board Statement:** Not applicable.

**Informed Consent Statement:** Not applicable.

**Data Availability Statement:** Data is not publicly available.

**Acknowledgments:** Novo Nordisk A/S, Cytiva, and the Strategic Innovation Program Process Industry IT and Automation (PiiA) are gratefully acknowledged for their financial support.

**Conflicts of Interest:** The authors declare that no commercial or financial conflicts of interest exist with regards to this work.

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
