# Peer review of "Automation of Modeling and Calibration of Integrated Preparative Protein Chromatography Systems"

_processes, doi:10.3390/pr10050945_

Round 1

Reviewer 1 Report

  1. Introduction

Although well written, this section should be reviewed as presented information is somewhat repetitive and a bit extensive for general readers.

  1. Theory

Mathematical models (equations) are numbered at right, however the numbering is hardly being used in text, except at line 363. Please review, by introducing equation numbers within text accordingly or deleting.

Line 87 -> sentence is ended with a comma? Please review;

Line 94 -> for clarity, equation should come after its description, at line 97/98, please review;

Line 99 -> no end point after "properties", please correct or review;

Line 101-> for clarity, equation should come after its description, at line 105/106, please review;

Line 109 -> for clarity, equation should come after its description, at line 113/114, please review;

Lines 81 and 115 -> parameterized, reparametrized -> please standardize throughout text;

Line 120 -> what is H parameter? -> please describe it in text;

Line 140 -> for clarity, equation should come after its description, after "GH.", please review;

  1. Materials and Methods

Line 160 -> "was used wich consisted"-> review and correct please;

Table 1 -> why "Three linear gradients" naming for Calibration step III? review or explain it at this section of the manuscript, please;

Line 180 -> protein sample concentration? 1M? -> mention it please;

Line 187 -> sample concentration? mention it please;

Line 220 -> no comma after "elution";

Sample concentrations used in experiments -> all experiments used concentrations as indicated in lines 152-153? -> Haven´t you performed dilutions? -> explain and clarify this for all experiments presented, please;

Column size/dimensions of HiTrap used are not mentioned in text -> models would not have to take in account column length? -> explain this, please;

  1. Results

Figure 2, 3 and 4 -> Figure results barely mentioned in text -> results shown are not described in text, which should be further explored, please review;

For instance, how good are your calibration/simulation chromatograms comparing to experimental ones? This is not presented or discussed at all. This point is somehow addressed by authors in general discussion section at lines 395-398, but this information could be more connected to the results presented.

Model Calibration results are column matrix independent? Have you tested IEX matrices other than agarose, for instance?  -> explain this, please; it seems that the issue above has been indirectly addressed by authors at line 303, but it would be nice if it could clearer for the readers.

  1. Conclusions

This section should be reviewed as it is just an overview and no conclusion is stated.

Reviewer 2 Report

Please see the attached document
